# Efficiently Disentangle Causal Representations

Yuanpeng Li,* Joel Hestness, Mohamed Elhoseiny, Liang Zhao & Kenneth Church

This paper proposes an efficient approach to learning disentangled representations with causal mechanisms based on the difference of conditional probabilities in original and new distributions. We approximate the difference with models' generalization abilities so that it fits in the standard machine learning framework and can be computed efficiently. In contrast to the state-of-the-art approach, which relies on the learner's adaptation speed to new distribution, the proposed approach only requires evaluating the model's generalization ability. We provide a theoretical explanation for the advantage of the proposed method, and our experiments show that the proposed technique is 1.9–11.0$\times$ more sample efficient and 9.4–32.4$\times$ quicker than the previous method on various tasks. The source code is available at `https://github.com/yuanpeng16/EDCR`.

## 1. Introduction

Causal reasoning is a fundamental tool that has shown significant impact in different disciplines [1–4], and it has roots in work by David Hume in the eighteenth century [5] and classical AI [6]. Causality has been mainly studied from a statistical perspective [7–10] with Judea Pearl's work on the causal calculus leading its statistical development.

More recently, there has been a growing interest in integrating statistical techniques into machine learning to leverage their benefits. Welling raises a particular question about how to disentangle correlation from causation in machine learning settings to take advantage of the sample efficiency and generalization abilities of causal reasoning [11]. Although machine learning has achieved important results on a variety of tasks like computer vision and games over the past decade (e.g., [12–15]), current approaches can struggle to generalize when the test data distribution is much different from the training distribution (common in real applications). Further, these successful methods are typically "data-hungry", requiring an abundance of labeled examples to perform well across data distributions. In statistical settings, encoding the causal structure in models has been shown to have significant efficiency advantages.

In support of the advantages of encoding causal mechanisms, [16] recently introduced an approach to disentangling causal representations in end-to-end machine learning by comparing the adaptation speeds of separate models that encode different causal structures. With this as the baseline, in this paper, we propose a more efficient approach to learning disentangled representations with causal mechanisms based on the difference of conditional probabilities in original and new distributions. The key idea is to approximate the difference with models' generalization abilities so that it fits in the standard machine learning framework and can be efficiently computed. In contrast to the state-of-the-art baseline approach, which relies on the learner's adaptation speed to new distribution, the proposed approach only requires evaluating the generalization ability of models.

Our method is based on the same assumption as the baseline that the conditional distribution $P(B|A)$ does not change between the train and transfer distribution. This assumption can be explained with an atmospheric physics example of learning $P(A, B)$, where $A$ (**Altitude**) causes $B$ (**Temperature**) [8]. The marginal probability of $A$ can be changed depending on, for instance, the place (Switzerland to a less mountainous country like the Netherlands), but $P(B|A)$ as the underlying causality mechanism does not change. Therefore, we can infer the causality from the robustness of predictive models on out-of-distribution [8, 17] and further learn causal representations (Section 2.4).

---

*Corresponding author: `yuanpeng16@gmail.com`

First Conference on Parsimony and Learning (CPAL 2024).

The proposed method improves efficiency by omitting the adaptation process and is more robust when the marginal distribution is complicated. We provide a theoretical explanation and experimental verification for the advantage of the proposed method. Our experiments show the proposed technique is 1.9–11.0× more sample efficient and 9.4–32.4× quicker than measuring adaptation speed on various tasks. We also argue that the proposed approach reduces the required hyperparameters, and it is straightforward to implement the approach within the standard machine learning workflows.

Our contributions can be summarized as follows.

- We propose an efficient approach to disentangling representations for causal mechanisms by measuring generalization.

- We theoretically prove that the proposed estimators can identify the causal direction and disentangle causal mechanisms.

- We empirically show that the proposed approach is significantly quicker and more sample-efficient for various tasks. Sample efficiency is important when the data size in transfer distribution is small.

## 2. Approach

To begin, we reflect on the tasks and the disentangling approach (as baseline) described by previous work [16]. The invariance of conditional distribution for the correct causal direction $P(B|A)$ is the critical assumption in the work, and we also follow it in this work. We notice that the baseline approach compares the adaptation speed of models on a transfer data distribution and hence requires significant time for adaptation. We propose an approach to learn causality mechanisms by directly measuring the changes in conditional probabilities before and after intervention for both $P(B|A)$ and $P(A|B)$. Further, we optimize the proposed approach to use generalization loss rather than a divergence metric—because loss can be directly measured in standard machine learning workflows—and we show that it is likely to predict the causal direction and disentangle causal mechanisms correctly.

### 2.1. Causality Direction Prediction

We start with the binary classification task as the first step towards learning disentangled representations for causal mechanisms. Given two discrete variables $A$ and $B$, we want to determine whether $A$ causes $B$ or vice-versa. We assume noiseless dynamics, and $A$ and $B$ do not have hidden confounders.

The training (transfer) data contains samples $(a, b)$ from training (transfer) distribution, $P_1$ ($P_2$). The baseline approach defines models that factor the joint distribution $P(A, B)$ into two causality directions $P_{A \to B}(A, B) = P_{A \to B}(B|A)P_{A \to B}(A)$ and $P_{B \to A}(A, B) = P_{B \to A}(A|B)P_{B \to A}(B)$. It then compares their speed of adaptation to transfer distribution.

Intuitively, the factorization with the correct causality direction should adapt more quickly to the transfer distribution. Suppose $A \to B$ is the ground-truth causal direction. For the correct factorization, conditional distribution $P_{A \to B}(B|A)$ is assumed not to change between the train and transfer distributions so that only the marginal $P_{A \to B}(A)$ needs adaptation. In contrast, for the factorization with incorrect causality direction, both the conditional $P_{B \to A}(A|B)$ and marginal distributions $P_{B \to A}(B)$ need adaptation. It is analyzed that updating a marginal distribution $P(A)_{A \to B}$ is likely to have lower sample complexity than the conditional distribution $P_{B \to A}(A|B)$ (only a part of $P_{B \to A}(A, B)$) because the latter has more parameters. Therefore, the model with correct factorization will adapt more quickly, and causality direction can be predicted from adaptation speed.

To leverage this observation, the baseline method defines a meta-learning objective. Let $\mathcal{L}_{A \to B}$ and $\mathcal{L}_{B \to A}$ be the log-likelihood of $P_{A \to B}(A, B)$ and $P_{B \to A}(A, B)$, respectively. It optimizes a regret $\mathcal{R}$

to acquire an indicator variable $\gamma \in \mathbb{R}$. If $\gamma > 0$, the prediction is $A \rightarrow B$, otherwise $B \rightarrow A$.

$$\mathcal{R} = -\log[\sigma(\gamma)\mathcal{L}_{A \rightarrow B} + (1 - \sigma(\gamma))\mathcal{L}_{B \rightarrow A}] \tag{1}$$

$\sigma$ is a sigmoid function $\sigma(x) = 1/(1 + \exp(-x))$.

## 2.2. Proposed Mechanism: Observe the Model's Conditional Distribution Divergence

Rather than relying on gradients of end-to-end predictions to identify the causal direction, we propose to directly observe the divergence of each model's conditional distribution predictions under the intervention of the transfer dataset.

We consider the distribution of a single model's predictions on the training and transfer distributions. These distributions depend on both the marginal and conditional distributions learned by the models. It is expected that a model encoding the correct causal direction will have learned a correct conditional distribution, but the marginal distribution might shift from the training to transfer datasets. Hence, we would like to ignore changes in the model's predictions that are due to marginal distribution differences and focus directly on the conditional distribution.

Given access to the model's conditional distribution predictions, we could use the KL-divergence to directly measure the conditional distribution differences between the training and transfer datasets. A model that encodes the correct causal structure should show no change in its conditional distribution predictions, so the divergence of these predictions should be slight. On the other hand, a model that encodes the incorrect causal structure is likely to show a significant divergence. We tie these observations together in the following proposition:

**Proposition 1.** *Given two data distributions with the same directed causality between two random variables, $A$ and $B$, the difference of the KL-divergences of their conditional distributions is an unbiased estimator of the correct causality direction between $A$ and $B$.*

This approach shares the same intuition as the baseline approach. The model with correct causal direction should only witness minimal changes in the conditional distribution of its predictions between train and transfer distributions. Thus, the KL-divergence of the predictions should not change for models with correct causal structure. The proof for Proposition 1 is included in Appendix A.

## 2.3. Proposed Simplification: Generalization Loss Approximates the Divergence

Although we can use the KL-divergence of the conditional probabilities of two data distributions as an unbiased estimator of causality direction, it requires additional computation to model conditional distribution in transfer distribution. However, in many practical end-to-end learning settings, it is more efficient and straightforward to acquire generalization loss, such as the cross-entropy, that can be used to approximate the conditional KL-divergence.

More specifically, we compare the generalization gaps between two causal models to approximate the causality direction. Let the generalization gap, $\mathcal{G}$, be the difference of a model's losses on the training and transfer datasets: $\mathcal{G}_{A \rightarrow B} = \mathcal{L}_{A \rightarrow B}^{\text{transfer}} - \mathcal{L}_{A \rightarrow B}^{\text{train}}$. Here, $\mathcal{L}_{A \rightarrow B}$ is the loss on the specified set. Further, we define a directionality score as the difference of generalization gaps:

$$\mathcal{S}_{\mathcal{G}} = \mathcal{G}_{B \rightarrow A} - \mathcal{G}_{A \rightarrow B}$$

We show that for an appropriately chosen loss, such as cross-entropy, $\mathcal{S}_{\mathcal{G}}$ is a biased but reasonable estimator of the correct causal direction. When $\mathcal{S}_{\mathcal{G}} > 0$, $A \rightarrow B$ is likely the correct causal direction– the generalization gap for the incorrect causal model dominates the score. We formalize this notion in Proposition 2 in Appendix B:

**Proposition 2.** *Given two data distributions with the same directed causality between two random variables, $A$ and $B$, the difference of their generalization gaps is a biased estimator of the correct causality direction between $A$ and $B$.*

In practice, for the tasks we test in the next section, we find that although $\mathcal{S}_\mathcal{G}$ is biased, it always indicates the correct causal direction. An intuitive understanding is that this approach measures how well models trained on train distribution can predict in transfer distribution–their generalization ability. Algorithm 1 summarizes the process for identifying a correct causal model. Appendix C describes that the generalization-based approach should converge more quickly than a gradient-based approach, especially in practical settings.

---

**Algorithm 1** The proposed approach for causality direction prediction.

1: Train $f_{A \to B}$ on training data, and get train loss $\mathcal{L}^{\text{train}}_{A \to B}$.
2: Train $f_{B \to A}$ on training data, and get train loss $\mathcal{L}^{\text{train}}_{B \to A}$.
3: Get transfer loss $\mathcal{L}^{\text{transfer}}_{A \to B}$ with $f_{A \to B}$ on transfer data.
4: Get transfer loss $\mathcal{L}^{\text{transfer}}_{B \to A}$ with $f_{B \to A}$ on transfer data.
5: Get generalization loss $\mathcal{G}_{A \to B} = \mathcal{L}^{\text{transfer}}_{A \to B} - \mathcal{L}^{\text{train}}_{A \to B}$.
6: Get generalization loss $\mathcal{G}_{B \to A} = \mathcal{L}^{\text{transfer}}_{B \to A} - \mathcal{L}^{\text{train}}_{B \to A}$.
7: Compute $\mathcal{S}_\mathcal{G} = \mathcal{G}_{B \to A} - \mathcal{G}_{A \to B}$.
8: If $\mathcal{S}_\mathcal{G} > 0$ return $A \to B$, else return return $B \to A$.

---

The baseline and proposed approaches are based on the same intuition of stable conditional distributions for correct causality direction. However, the baseline approach must observe gradients during transfer distribution training, while the proposed approach looks directly at the changes in model outputs on the transfer data distribution. Informally, we note that these approaches are roughly equivalent by observing the following: for the correct causal direction,

$$\mathcal{G}_{\cdot \to \cdot} = \mathcal{L}^{\text{transfer}}_{\cdot \to \cdot} - \mathcal{L}^{\text{train}}_{\cdot \to \cdot} \qquad\qquad \nabla \mathcal{G}_{\cdot \to \cdot} = \nabla \mathcal{L}^{\text{transfer}}_{\cdot \to \cdot} - \nabla \mathcal{L}^{\text{train}}_{\cdot \to \cdot}$$

Since $\nabla \mathcal{L}^{\text{train}} = 0$, $\nabla \mathcal{G}_{\cdot \to \cdot} = \nabla \mathcal{L}^{\text{transfer}}_{\cdot \to \cdot}$. The previous work shows that $\nabla_{\theta_i} \mathcal{L}^{\text{transfer}}_{\cdot \to \cdot} = 0$ for model parameters, $\theta_i$, representing correct causal structures (i.e., joint distribution), so $\nabla \mathcal{L}^{\text{transfer}}_{\cdot \to \cdot}$ should also be small. Similar to that, we expect $\mathcal{G}_{\cdot \to \cdot}$ to be small relative to the incorrect causal model.

## 2.4. Representation Learning

So far, we assume that the causal variables are already disentangled in the raw inputs. We are interested in extending it to situations where the variables are entangled in input, such as pixels or sounds. Here, we want to learn representations that disentangle causal variables.

Following the previous work, we suppose the true causal variables $(A, B)$ generate input observation $(X, Y)$ with ground truth decoder, $\mathcal{D}$. We want to train an encoder, $\mathcal{E}$, that converts input $(X, Y)$ to a hidden representation, $(U, V)$. The decoder and encoder are rotation matrices.

$$\begin{bmatrix} X \\ Y \end{bmatrix} = R(\theta_\mathcal{D}) \begin{bmatrix} A \\ B \end{bmatrix} \qquad\qquad \begin{bmatrix} U \\ V \end{bmatrix} = R(\theta_\mathcal{E}) \begin{bmatrix} X \\ Y \end{bmatrix}$$

Then, we treat $(U, V)$ as observed input and build causal modules on them in a way similar to the causality direction prediction problem (Section 2.1). If the encoder is learned correctly, we expect to obtain $(U, V) = (A, B)$ with $\theta_\mathcal{E} = -\theta_\mathcal{D}$, or $(U, V) = (B, A)$ with $\theta_\mathcal{E} = \theta_\mathcal{D}$. The baseline approach addresses this problem with the loss function in Equation (1) by extending meta-parameters to encoder parameters in the meta-learning framework.

We apply our method to learn representation without using the adaptation process. Suppose the conditional distributions $P(V|U)$ is modeled by $f_{U \to V}$, and $P(U|V)$ by $f_{V \to U}$. Also, the original losses are $\mathcal{L}_{U \to V}$ for $f_{U \to V}$ and $\mathcal{L}_{V \to U}$ for $f_{V \to U}$ on train data. We optimize the following objective.

$$\mathcal{L} = \mathcal{L}_{U \to V} + \mathcal{L}_{V \to U} + \lambda \min\{\mathcal{G}_{U \to V}, \mathcal{G}_{V \to U}\}$$

$\lambda$ is a hyperparameter for interpolation. Note that we compute $\mathcal{L}_{U \to V}, \mathcal{L}_{V \to U}$ with train data, and $\mathcal{G}_{U \to V}, \mathcal{G}_{V \to U}$ with transfer data. When we use mini-batches, this means that there are two types

---

**Algorithm 2** The proposed approach for representation learning.

---

1:  Initialize all parameters.
2:  **for** each iteration **do**
3:      Update predictor parameters with train samples, and loss $\mathcal{L}_{U \to V} + \mathcal{L}_{V \to U}$.
4:      Update encoder parameters with transfer samples, and loss $\lambda \min\{\mathcal{G}_{U \to V}, \mathcal{G}_{V \to U}\}$.
5:  **end for**

---

of mini-batches for data from train and transfer datasets, respectively. Train data is used to learn $f_{U \to V}$ and $f_{V \to U}$, and transfer data is used to learn encoder $\theta_{\mathcal{E}}$. Please see Algorithm 2 for details.

An intuitive understanding is that we want to learn a representation that works well in both causal directions in train distribution and at least one causal direction in transfer data distribution. Therefore, the models are trained well in train distribution, and the representation recovers causal variables because it works on both distributions in at least one causality direction.

## 3. Experiments

We run experiments to show that predicting the causal direction and disentangling causal representations can be more efficient with the proposed approach. For a fair comparison, we follow the same experimental setup as the baseline [16].

### 3.1. Causality Direction Prediction

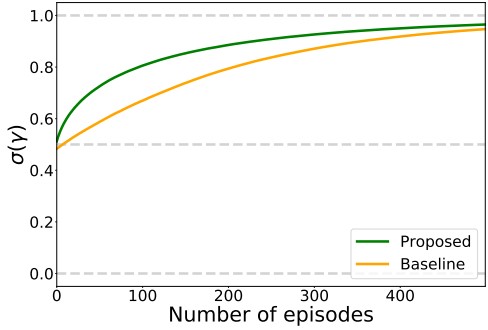

(a) $\sigma(\gamma)$ (vertical) on transfer data with the number of episodes (horizontal).

(b) Accuracy (vertical) on transfer data with the number of episodes (horizontal).

Figure 1: Experiments to evaluate efficiency in causality direction prediction. The model is discrete, and both $A$ and $B$ are ten-dimensional variables ($N = 10$). The proposed approach is green. The baseline approach [16] is orange.

We first compare the approaches in the same meta-learning setting. For the proposed approach, we replace the log-likelihood of joint probabilities with the generalization loss. The overall loss for the proposed approach $\mathcal{R}'$ is similar to Equation (1) with $\mathcal{L}_{A \to B} = \mathcal{G}_{A \to B}$ and $\mathcal{L}_{B \to A} = \mathcal{G}_{B \to A}$.

$$\mathcal{R}' = -\log[\sigma(\gamma)\mathcal{G}_{A \to B} + (1 - \sigma(\gamma))\mathcal{G}_{B \to A}]$$

The first case we consider is the discrete model with both $A$ and $B$ being ten-dimensional variables ($N = 10$). For other cases, please refer to Appendix E. The model architecture, the same as in the baseline, has four modules: $P_{A \to B}(A), P_{A \to B}(B|A), P_{B \to A}(B), P_{B \to A}(A|B)$. The baseline uses all of them, and the proposed approach only uses the conditionals. We parameterize these probabilities via Softmax of unnormalized tabular quantities and train the modules independently with a maximum likelihood estimator. We then predict the causality direction on transfer data with each approach.

For each approach, we fix the ground-truth causal direction as $A \rightarrow B$, and run 100 experiments with different random seeds. We plot the mean value of $\sigma(\gamma)$ in Figure 1a. The proposed method (green) is above the baseline method (orange). This comparison shows that using generalization loss is more efficient than the log-likelihood of joint probabilities. Especially in the first several steps, the proposed approach moves toward the correct $\gamma$ more quickly than the baseline approach. This quicker convergence is important when the transfer data is expensive. We also observe that for the baseline method, $\sigma(\gamma)$ is less than 0.5 for the first several steps, which might be because the factorization with incorrect causality direction has more parameters to be updated, leading to faster adaptation.

Since the proposed approach does not need the adaptation process, it does not need to update $\sigma(\gamma)$ with gradients. Instead, we can calculate a score, $\mathcal{S}_{\mathcal{G}} = \mathcal{G}_{B \rightarrow A} - \mathcal{G}_{A \rightarrow B}$, to detect causality direction. This score is not directly comparable with $\sigma(\gamma)$ in the baseline approach because they have different scales and semantics. To compare them, we reformulate the causality direction prediction problem as a binary classification problem and evaluate the accuracy of the classification. For each approach, we run 100 experiments with the same setting as above. We then compute the accuracy as the number of experiments with successful prediction over all experiments. For the baseline approach, we count an experiment as successful when $\gamma > 0$. For the proposed approach, we count when $\mathcal{S}_{\mathcal{G}} > 0$.

We plot accuracy along with the number of iterations to compare sample efficiency as shown in Figure 1b. The experimental result demonstrates that the proposed approach (green) achieves 100% accuracy after just 3 iterations with 5.1 ms, while the baseline approach (orange) takes 33 iterations to achieve 100% accuracy with 165.1 ms.

Since our method only requires estimating a scale value of loss from transfer data, it is more sample efficient than the original method that requires learning/updating all parameters in a model during adaptation. The experiments show that our method is 11.0 times sample efficient and 32.4 times quicker compared with the baseline approach. Also, the baseline accuracy for the first several steps is less than 50%. This is consistent with the previous observation that the $\sigma(\gamma) < 0.5$ for the first several steps.

## 3.2. Representation Learning

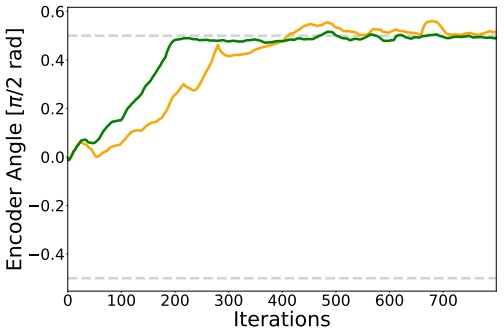
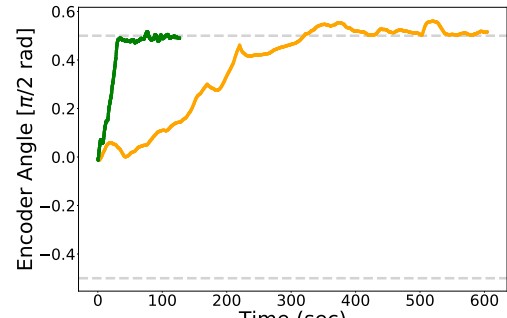

(a) $\theta$ values ($\times \pi/2$) with number of iterations.

(b) $\theta$ values ($\times \pi/2$) with computation time in seconds.

Figure 2: Experiments to evaluate efficiency in representation learning with the same setting. The curve of the proposed approach (green) is above that of the baseline approach (orange), indicating the proposed method has both better sample and time complexity.

We evaluate representation learning in the same setting as in previous work. We hope finding the correct causality direction helps to learn causal representations. We set the correct causal direction as $A \rightarrow B$. With a sample $(A, B)$, a fixed decoder $\mathcal{D} = R(\theta_{\mathcal{D}})$ converts it to an observation $(X, Y)$,

where $R$ is a rotation matrix. We then use both approaches to learn encoders $\mathcal{E} = R(\theta_\mathcal{E})$ that map $(X, Y)$ to representation $(U, V)$. We set decoder parameter $\theta_\mathcal{D} = -\pi/4$, so that the expected encoder parameter should be $\pi/4$ or $-\pi/4$.

For both approaches, we use the same experiment setting and hyperparameters. In each iteration, the baseline approach uses transfer samples to adapt the model by learning gradient descent for several steps and obtain regret to update the encoder. On the other hand, we directly use loss instead of analyzing with a statistical test because this can be neurally integrated with neural network training for disentangled representation learning.

We plot the relation of $\theta$ values along with the number of iterations and time to compare sample and time efficiency. The result for sample efficiency is in Figure 2a. The proposed approach (green) reaches near $\pi/4$ after around 220 iterations. Baseline approach (orange) reaches near $\pi/4$ after around 410 iterations. The proposed approach is around 1.9 times sample efficient compared with the baseline approach. The result for time efficiency is in Figure 2b. The proposed approach (green) reaches near $\pi/4$ in around 34 seconds. Baseline approach (orange) reaches near $\pi/4$ in around 319 seconds. The proposed approach is 9.4 times quicker than the baseline approach.

The results show that the proposed approach is both sample and time-efficient for the representation learning task. Especially, the time efficiency shows the proposed method's advantage that it does not need multiple steps of adaptation in each iteration. It even does not need to compute gradient with backpropagation or update model parameters.

## 4. Discussions

To further understand the proposed approach, we discuss alternative metrics for distribution differences, the proposed approach's working conditions, and the baseline. We also extend experiments on real data and neural network models.

**Alternative Metrics**  The proposed approach may even work when using other metrics that are easily accessed in standard machine learning workflows. Many loss functions share properties of distance metrics and smoothness conditions that make them helpful in comparing the generalization ability of different models. Such metrics may be biased but can also be used to assess causality direction. We describe and experiment with metrics such as the KL-divergence loss and the gradient norm in Appendix F. Empirical tests show that both alternative metrics can identify causality direction in the prediction task. One may choose these alternative metrics depending on the workflow.

**Robustness**  We compare the proposed and the baseline approach on the working conditions. One potential problem of the baseline approach is the dependency on marginal distributions. This not only requires more samples and computation, but when the marginal distribution for the correct factorization $P(A)$ is complicated, it may produce incorrect causality predictions (suppose $A \rightarrow B$ is the correct causality direction). We show an example of such a case in Appendix G. We design a marginal distribution of cause variable $P(A)$ with a complicated hidden structure, which makes it harder to adapt after intervention than the marginal distribution of effect variable $P(B)$, and conditional distribution $P(A|B)$. We use a mixture model for $P(A)$ with $K$ dimensional hidden variables $C$ and do not change models for other distributions. The result shows that the proposed approach is more robust than the baseline one in this case.

**Broader Situations and Real Data**  In addition to comparing the proposed approach with the previous work in the same setting to understand the fundamental mechanism for causality learning, we also evaluate whether the proposed method extends naturally to broader situations. We use real data of temperature and altitude [18]. We scale the data to have unit variance. We apply the proposed approach with both linear models and neural networks. We also add noise to the data and observe the changes in performance. The details can be found in Appendix H. The results show that the proposed approach works for the real data and applies to both linear models and neural

networks. It resists some level of noise (standard deviation $\leq 1$), but not when the noise is too large.

# 5. Related Work

Our work is closely related to causal reasoning, disentangled representation learning, and its application to domain adaptation and transfer learning.

**Causal Reasoning** There is rich literature on causal reasoning based on the pre-conceived formal theory since Pearl's seminal work on do-calculus [7, 19]. The major approach is structure learning in Bayesian networks. Discrete search and simulated annealing are reviewed in [20]. Minimum Description Length (MDL) principles are commonly used to score and search [21, 22]. Bayesian Information Criterion (BIC) is also used to search when models have high relative posterior probability [20].

Prior work uses observation but does not consider interventions and focuses on likelihood learning or hypothesis equivalence classes [20]. Later, approaches have also been proposed to infer the causal direction only from observation [17], based on not generally robust assumptions on the underlying causal graph. The impact of interventions on probabilistic graphical models is also introduced in [23]. Another early work [8] uses intervention to detect causal direction, but it does not address representation learning, which is an important problem in the modern neural network and the main topic of this paper. Further, our method should be more efficient than the fast approximation method [8], which takes iterations to fit a model. In contrast, we only run one forward pass on a fixed model.

Recently, a meta-learning approach is adopted to draw causal inferences from purely observational data. [24] focuses on training an agent to learn causal reasoning by meta reinforcement learning; [16] focuses on predicting causal structure based on how fast a learner adapts to new distributions. In contrast, we propose an efficient approach for disentangling causal mechanisms based on the difference of conditional probabilities in original and new distributions, assuming causal mechanisms hold in both distributions. More recently, generative models have been augmented with causal capabilities to learn to generate images and also to plan [25, 26].

**Disentangled Representation Learning** This work discovers the underlying causal variables and their dependencies. This is related to disentangling variables [27] and disentangled representation learning [28, 29]. Some work points out that assumptions, including priors or biases, should be necessary to discover the underlying explanatory variables [27, 30]. Another work [30] reviews and evaluates disentangling and discusses different metrics.

A strong assumption of disentangling is that the underlying explanatory variables are marginally independent of each other. Many deep generative models [31] and independent component analysis models [28, 32, 33] are built on this assumption. However, in realistic cases, this assumption is often not likely to hold. Some recent work addresses compositionality learning [34, 35] without such assumption, but they do not address how to use causal mechanisms for representation learning.

**Domain Adaptation and Transfer Learning** Previous work finds a subset of features to best predict a variable of interest [36]. However, the work is about feature selection, and our work is about causality direction and representation learning.

Other work also examines incorrect inference and turns the problem into a domain adaptation problem [37]. Counterfactual regression is proposed to estimate each effect from observation [38]. Another approach is to find a subset that makes the target independent from the variable selection [39]. To do that, they assume the invariance of conditional distribution in train and target domains.

Also, competition may be introduced to recover a set of independent causal mechanisms, which leads to specialization [40].

# 6. Conclusion

In this paper, we propose an efficient simplification of a technique to disentangle causal representations. Unlike the baseline approach, which requires significant adaptation time and sophisticated comparisons of the models' adaptation speed, the proposed approach directly measures generalization ability based on the divergence of conditional probabilities from the original to transfer data distributions. We provide a theoretical explanation for the advantage of the proposed method, and our experiments demonstrate that the technique is significantly more efficient than the approach based on adaptation speed in causality direction prediction and representation learning tasks. We hope this work in causal representation learning will be helpful for more advanced artificial intelligence.

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

# A. Proof that KL-divergence Can Be Used to Identify Causality Direction

Here, we prove that we could use conditional KL-divergence to detect causality direction.

**Proposition 1.** *Given two data distributions with the same directed causality (e.g., $A \to B$) between two random variables, $A$ and $B$, the difference of the KL-divergences of their conditional distributions (i.e., $P(B|A)$) is an unbiased estimator of the correct causality direction.*

*Proof.* First, let $P_1$ and $P_2$ be the distributions such that $P_1 \neq P_2$, but their conditional distributions are the same. Let $D_{\mathrm{KL}}(\cdot \| \cdot)$ be conditional KL-divergence. $\mathrm{CrossEntropy}_{A \to B}(P_i, P_j)$ is the conditional cross-entropy, $-\sum_{\forall(a,b)} P_i(a,b) \log P_j(b|a) = \mathbb{E}_{A,B \sim P_i}[-\log P_j(B|A)]$. Let $H_{A \to B}(\cdot)$ be conditional entropy.

By the KraftMcMillan theorem, we have

$$
\begin{aligned}
D_{A \to B} &= D_{\mathrm{KL}}(P_2(B|A) \| P_1(B|A)) \\
&= \mathbb{E}_{A,B \sim P_2}[\log P_2(B|A) - \log P_1(B|A)] \\
&= \mathbb{E}_{A,B \sim P_2}[-\log P_1(B|A)] - \mathbb{E}_{A,B \sim P_2}[-\log P_2(B|A)] \\
&= \mathrm{CrossEntropy}_{A \to B}(P_2, P_1) - H_{A \to B}(P_2)
\end{aligned}
$$

Without loss of generality, suppose $A \to B$ is the correct causality direction. Since $P_1(B|A) = P_2(B|A)$, we have that $\mathrm{CrossEntropy}_{A \to B}(P_2, P_1) = H_{A \to B}(P_2)$ and

$$
\begin{aligned}
D_{A \to B} &= \mathrm{CrossEntropy}_{A \to B}(P_2, P_1) - H_{A \to B}(P_2) \\
&= H_{A \to B}(P_2) - H_{A \to B}(P_2) \\
&= 0
\end{aligned}
$$

Finally, if we define a directionality score, $\mathcal{S}_{D_{\mathrm{KL}}}$, as the difference of KL-divergence of the two distributions, we can identify the incorrect causal direction by its greater divergence. Namely,

$$
\begin{aligned}
\mathcal{S}_{D_{\mathrm{KL}}} &= D_{B \to A} - D_{A \to B} \\
&= [\mathrm{CrossEntropy}_{B \to A}(P_2, P_1) - H_{B \to A}(P_2)] - 0 \\
&> 0 \implies A \to B
\end{aligned}
$$

Here we use the property that entropy is less than cross-entropy when the two distributions are not equal. $\qquad \square$

# B. Proof that Generalization Loss Closely Approximates Divergence

Here, we prove that we can replace conditional KL-divergence with a standard generalization loss calculation to approximate the causal directionality score, $\mathcal{S}_{D_{\mathrm{KL}}}$, in Proposition 1. This approximation introduces bias into the score, but we expect this bias to be slight relative to the difference of conditional KL-divergences of the incorrect causal model. Thus, it can be used as a good estimator of causality direction.

**Proposition 2.** *Given two different data distributions, $P_1$ and $P_2$, with the same directed causality between random variables, $A$ and $B$, consider the generalization loss defined as the conditional cross-entropy of a given distribution:*

$$
\begin{aligned}
\mathcal{L}_{A \to B}^{P_i} &= \mathrm{CrossEntropy}_{A \to B}(P_i, P_1) \\
&= \mathbb{E}_{A,B \sim P_i}[-\log P_1(B|A)]
\end{aligned}
$$

*And define the generalization gap as $\mathcal{G}_{\cdot \to \cdot} = \mathcal{L}_{\cdot \to \cdot}^{P_2} - \mathcal{L}_{\cdot \to \cdot}^{P_1}$. Finally, define the modified causality score, $\mathcal{S}_{\mathcal{G}} = \mathcal{G}_{B \to A} - \mathcal{G}_{A \to B}$. If $\mathcal{S}_{D_{\mathrm{KL}}} = D_{B \to A} - D_{A \to B}$ as defined in Proposition 1, then $\mathcal{S}_{\mathcal{G}} = \mathcal{S}_{D_{\mathrm{KL}}} - [\Delta H(B) - \Delta H(A)]$. $\mathcal{S}_{\mathcal{G}}$ is a biased estimator of which model has correct causality direction.*

*Proof.* Let $H(\cdot, \cdot)$ be the joint entropy. The generalization gap is

$$
\begin{aligned}
\mathcal{G}_{A \to B} &= \mathcal{L}_{A \to B}^{P_2} - \mathcal{L}_{A \to B}^{P_1} \\
&= \text{CrossEntropy}_{A \to B}(P_2, P_1) - H_{A \to B}(P_1) \\
&= [\text{CrossEntropy}_{A \to B}(P_2, P_1) - H_{A \to B}(P_2)] + H_{A \to B}(P_2) - H_{A \to B}(P_1) \\
&= D_{A \to B} + H_{A \to B}(P_2) - H_{A \to B}(P_1) \\
&= D_{A \to B} + H(P_2(B|A)) - H(P_1(B|A)) \\
&= D_{A \to B} + [H(P_2(A, B)) - H(P_2(A))] - [H(P_1(A, B)) - H(P_1(A))] \\
&= D_{A \to B} + [H(P_2(A, B)) - H(P_1(A, B))] - [H(P_2(A)) - H(P_1(A))] \\
&= D_{A \to B} + \Delta H(A, B) - \Delta H(A) \\
\mathcal{G}_{B \to A} &= D_{B \to A} + \Delta H(A, B) - \Delta H(B)
\end{aligned}
$$

Here $\Delta H(A, B)$, $\Delta H(A)$ and $\Delta H(B)$ are the change in entropy from $P_1$ to $P_2$ for $(A, B)$, $A$ and $B$, respectively. Further, the modified causality score is the difference of these generalization gaps:

$$
\begin{aligned}
\mathcal{S}_{\mathcal{G}} &= \mathcal{G}_{B \to A} - \mathcal{G}_{A \to B} \\
&= [D_{B \to A} + \Delta H(A, B) - \Delta H(B)] - [D_{A \to B} + \Delta H(A, B) - \Delta H(A)] \\
&= [D_{B \to A} - D_{A \to B}] - [\Delta H(B) - \Delta H(A)] \\
&= \mathcal{S}_{D_{\text{KL}}} - [\Delta H(B) - \Delta H(A)]
\end{aligned}
$$

Without loss of generality, suppose $A \to B$ is the correct causal direction, then $D_{A \to B} = 0$.

$$
\mathcal{S}_{\mathcal{G}} = D_{B \to A} - [\Delta H(B) - \Delta H(A)]
$$

When $\mathcal{S}_{\mathcal{G}} > 0$ (and $[\Delta H(B) - \Delta H(A)]$ is small), the prediction of causality direction is correct. □

Informally, we expect the bias term, $[\Delta H(B) - \Delta H(A)]$, to be small relative to the divergence, $D_{B \to A}$, in real application settings. If $A \to B$, then $A$ and $B$ will have some (positive or negative) correlation–they will vary in a correlated way under intervention on $A$. Unless the conditional distribution, $P_1(B|A) = P_2(B|A)$, contains large portions with high entropy, a change in $A$ will likely cause a commensurate change in $B$. Thus, it is likely that $|\Delta H(A)| \approx |\Delta H(B)|$, so it is unlikely that $\Delta H(A)$ and $\Delta H(B)$ are both large and have opposite sign.

Further, in cases where $\Delta H(A)$ is large, but $\Delta H(B)$ is small, it is frequently an indicator that the selected $P_i$ are "lucky" (uncommon) or the causality between $A$ and $B$ is weak. Consider the example in which $H(P_1(A))$ is small, but $H(P_1(B))$ is large. This case can occur if $H(P_1(B|A))$ is small, but it is unlikely because the (low-entropy) conditional distribution must impose the disorder on $P_1(B)$. It is easy to see that the reverse situation ($H(P_1(B|A))$ is large) implies that the causality between $A$ and $B$ is weak, such that causality direction prediction may be difficult anyway.

Finally, if we want to relax the assumption that either $A \to B$ or $B \to A$ (e.g., a null-hypothesis that $A$ and $B$ are not causally related), we could introduce a significance test to decide whether we can accept the selection of causal direction from either method. Such a significance test would need to be mindful of the magnitude of any biases introduced in the approach.

## C. Generalization Loss is Likely to be More Practical

Here, we explain how generalization loss is likely to be more practical than gradient-based approaches for identifying a model with the correct causal direction. Both the baseline and proposed approaches rely on the convergence of estimators toward a ground truth model, but gradient-based approaches require much stronger assumptions to guarantee convergence, and they are more susceptible to practical challenges like tuning optimizer hyperparameters.

First, both gradient- and generalization-based approaches have the same asymptotic convergence characteristics, with the prediction error proportional to $\frac{1}{\sqrt{n}}$, where $n$ is the number of steps or

samples. Many theorems state that gradient-based approaches can have asymptotic convergence to within an $\varepsilon$ error based on number of gradient descent steps, $k$: $\varepsilon = O(\frac{1}{\sqrt{k}})$. These proofs rely on many assumptions that cannot be guaranteed in real-world datasets, such as convexity, regularity, or smoothness of error surface, using optimal settings of optimization parameters for best-case step sizes, etc. [41–43]. These bounds can also tend to be loose for real data distributions.

On the other hand, non-parametric estimators, such as the sample average generalization loss proposed in this work, do not have a model, so their convergence rates can depend only on the structure of the dataset (e.g., first or second moments). For instance, early estimator convergence results show that after summarizing $n$ samples, the error in an estimator can be within $\varepsilon$ of the true value: $\varepsilon = O(\frac{1}{\sqrt{n}})$ [44, 45]. The constants in these bounds are just the moments that characterize the dataset (e.g., mean, variance). Like the gradient-based convergence rates, these non-parametric convergence rates can also be loose for real data distributions.

Thus, in practical settings, it is likely that a generalization-based approach (non-parametric estimators) will converge more quickly on causality identification in real-world datasets. The proposed approach does not require further model optimization (fine-tuning), so it is not susceptible to poorly chosen hyperparameters. Further, it is unlikely to encounter complex error surface topology. Thus, in practice, it is likely to converge more quickly than gradient-based causality identification techniques.

## D. Representation Learning

We run representation learning experiments for both approaches on a GeForce GTX TITAN Z with PyTorch. We use the original implementation for the baseline approach and extend it for the proposed approach. The hyperparameters are the same as the original setting [16].

## E. More Experiments on Causality Direction Predictions

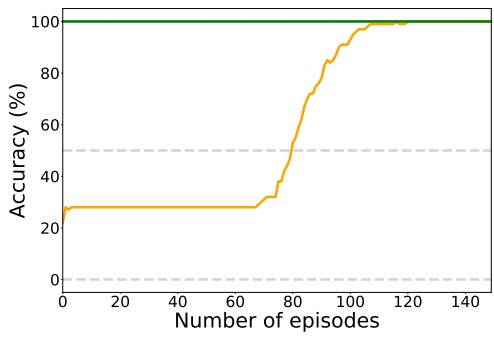

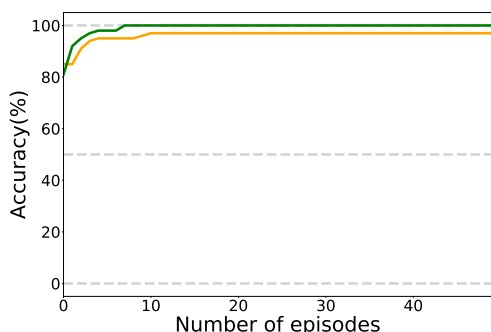

(a) $\sigma(\gamma)$ (vertical) on transfer data along with the number of episodes (horizontal).

(b) Accuracy (vertical) on transfer data with computation time in seconds (horizontal).

Figure 3: Experiments to evaluate efficiency in causality direction prediction. The model is discrete, and both $A$ and $B$ are 100-dimensional variables ($N = 100$).

We would like to extend causality direction prediction experiments to high dimensional variables and continuous variables. We still follow the experiment settings in previous work [16] and compare the proposed and the baseline approaches. We run all the experiments of causality direction prediction with Intel Core i9 9920X 12-Core Processors.

For high dimensional variables, we run experiments with $N = 100$. We use the same setting as the experiments with $N = 10$ but only changed the $N$. The result for sample efficiency is in Figure 3a.

Similar to the case of $N = 10$, the proposed approach (green) is significantly more efficient than the baseline approach (orange) in both sample.

For continuous variables, we used the same setting as the previous work [16]. For the proposed method, we only use one sample for each episode. The plot shows that the proposed approach (green) is more efficient than the baseline approach (orange) in sample number (Figure 3b).

## F. Alternative Metrics

The proposed approach may even work when using other metrics that are easily accessed in standard machine learning workflows. Many loss functions share properties of distance metrics and smoothness conditions that make them helpful in comparing the generalization ability of different models. Such metrics may be biased but can also be used to assess causality direction. We describe and experiment with metrics such as the KL-divergence loss and the gradient norm. Empirical tests show that both alternative metrics can identify causality direction in the prediction task. One may choose these alternative metrics depending on one's workflow.

For KL-divergence, we define the metric as follows and show an example result in Figure 4a.

$$\mathcal{S}_{D_{\text{KL}}} = D_{\text{KL}}[P_2(B|A)\|P_1(B|A)] - D_{\text{KL}}[P_2(A|B)\|P_1(A|B)]$$

For gradient $L_2$ norm, we define the metric as the follows, and we show an example result in Figure 4b.

$$\mathcal{S}_{L_2} = L_2\Big(\frac{\partial \mathcal{L}_{A \to B}^{\text{transfer}}}{\partial \theta_{A \to B}}\Big) - L_2\Big(\frac{\partial \mathcal{L}_{B \to A}^{\text{transfer}}}{\partial \theta_{B \to A}}\Big)$$

Note that the model or optimization algorithm does not have special requirements on the gradient, such as gradient clipping.

These results indicate that other loss functions can behave similarly to the conditional cross-entropy when identifying causality direction. The initial samples already show a large gap between the correct and incorrect causal models, and after just a few tens of examples, the difference is close to convergence.

We expect that the same intuition applies to these other loss metrics. In particular, these loss metrics both satisfy distance metric properties of non-negativity and the triangle inequality. Further, they have continuity and smoothness characteristics that suggest any biases they may have should behave like conditional cross-entropy. Our empirical results confirm these intuitions.

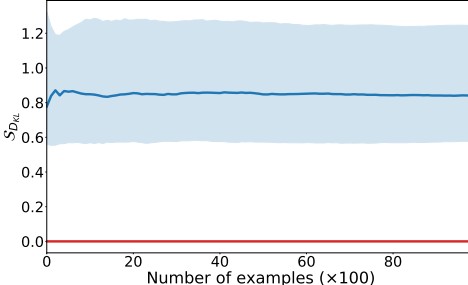

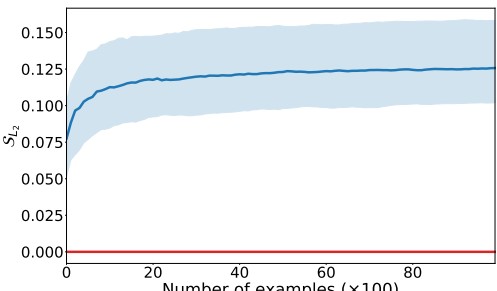

(a) Difference of KL-divergence $\mathcal{S}_{D_{\text{KL}}}$ along with the number of transfer samples in the proposed approach.

(b) Difference of $L_2$ gradient norm $\mathcal{S}_{L_2}$ along with the number of transfer samples in the proposed approach.

Figure 4: Experiments for other metrics. The model is discrete and $N = 10$. Curve (blue) is median over 100 runs, with 25-75% quantiles intervals, and it is significantly above zero (red), indicating these metrics are good indicators for causality learning.

# G. Robustness

To compare the robustness of the proposed approach with the baseline, we design $P(A)$ with a complicated hidden structure so that it is harder to adapt after intervention than $P(B)$ and $P(A|B)$. For $P(A)$, we use a mixture model with $K$ dimensional hidden variables $C$ and keep other distributions unchanged.

$$P(A; \theta, \phi) = \sum_{k=1}^{K} P(A|C_k; \phi)P(C_k; \theta)$$

where $\theta \in \mathbb{R}^K, \phi \in \mathbb{R}^{K \times N}$. We set $N = 10, M = 10, K = 200$. For the baseline approach, we plot the improvement of log-likelihood during adaptation along with the number of samples ($\times 100$) in Figure 5a. The model with correct causality direction (blue) adapts slower than that of incorrect direction (red), which means the fundamental intuition of baseline approach does not apply in this situation. For the proposed approach, we plot the difference of generalization losses $\mathcal{S}_{\mathcal{G}} = \mathcal{G}_{B \to A} - \mathcal{G}_{A \to B}$ along with the number of samples ($\times 100$) in Figure 5b. The difference (blue) is significantly larger than zero (red). This is natural because the proposed approach does not use $P(A)$ so that it is not influenced by whether $P(A)$ has a complicated hidden structure. This experiment shows that the proposed approach is more robust than the baseline one in this case.

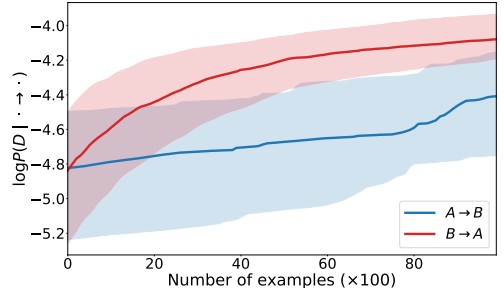
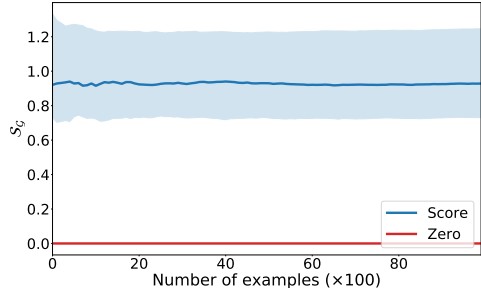

(a) Incorrect result from baseline approach. Log-likelihood during adaptation along with the number of samples ($\times 100$). The correct causal model (blue) adapts more slowly than that of the incorrect model (red).

(b) Result of the proposed approach. The difference of generalization losses $\mathcal{S}_{\mathcal{G}} = \mathcal{G}_{B \to A} - \mathcal{G}_{A \to B}$ along with the number of samples ($\times 100$). The difference (blue) is significantly larger than zero (red).

Figure 5: Experiments to evaluate robustness. $A \to B$ is the correct causality direction. Here, $N = 10, M = 10, K = 200$. Curves are median over 100 runs, with 25-75% quantiles intervals. The result shows that the models with correct causalities are slow to adapt in the baseline approach (a), which means the baseline approach does not work, but the proposed approach (b) works.

# H. Real Data Experiments

We use the altitude and temperature dataset [18], where altitude $(A)$ causes temperature $(B)$. We generate the train and the transfer datasets in the following way. We first divide the data into two sets, $D_1$ and $D_2$, according to their $A$ values. Samples in $D_1$ have larger or equal $A$ values than those in $D_2$. We respectively shuffle and separate $D_1$ and $D_2$ to two subsets with ratio of 9:1, generating $D_1^1, D_1^2, D_2^1, D_2^2$. We then merge sets to generate train data $(D_1^1 \cup D_2^2)$ and transfer data $(D_1^2 \cup D_2^1)$.

We use both a linear regression model with $L_2$ loss and a neural network model with two layers. The hidden layer has 100 nodes with ReLU activation. We use scikit-learn [46] for implementation. We run experiments 1,000 times with different random seeds, and the predicted causal directions are always correct.

For different noise levels, we use Gaussian noise with different standard deviations. In the altitude-temperature experiment with the linear model, the success rate starts to drop to $99.8 \pm 0.1\%$ at 0.5

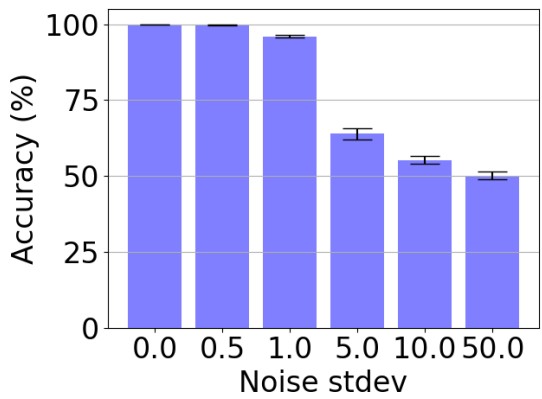

Figure 6: Accuracy when adding noise.

and becomes $96.0 \pm 0.4\%$ at 1.0, $63.9 \pm 1.8\%$ at 5.0, $55.3 \pm 1.2\%$ at 10.0, and $50.1 \pm 1.3\%$ at 50.0 (Figure 6). This means the method is robust to some noise (around 1 or less) but does not work when the noise is too large.

## I. Robustness and edge cases

We look into more details of the robustness and the edge cases for the difference in the generalization loss gap as a predictor. We first discuss the relation between the difference in KL divergence $\mathcal{S}_{D_{KL}}$ and the difference in loss gap $\mathcal{S}_{\mathcal{G}}$. We also discuss the relation between intervention and $\mathcal{S}_{\mathcal{G}}$. We then give a counterexample based on the analysis.

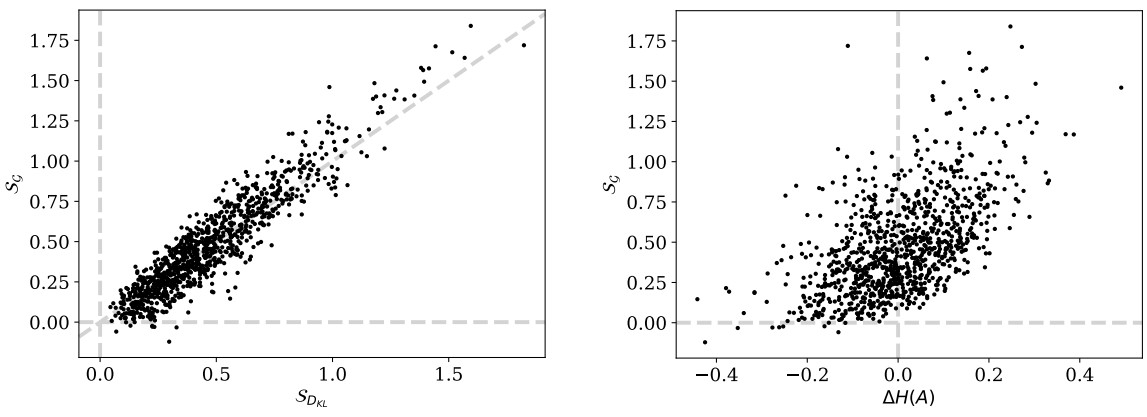

(a) The relation between $\mathcal{S}_{D_{KL}}$ and $\mathcal{S}_{\mathcal{G}}$. They are close to each other. When $\mathcal{S}_{\mathcal{G}}$ is negative, it is close to zero.

(b) The relation between $\mathcal{S}_{D_{KL}}$ and $\Delta H(A)$. $\mathcal{S}_{D_{KL}}$ can be negative when $\Delta H(A)$ is negative.

Figure 7: Experiments for the relation between statistics.

**Estimator** We look at the relation between the difference in generalization loss gap $\mathcal{S}_{\mathcal{G}}$ (proposed estimator) and the difference in KL divergence $\mathcal{S}_{D_{KL}}$. We assume $A \rightarrow B$, so $\mathcal{S}_{D_{KL}}$ is positive by Proposition 1. The estimation is correct if $\mathcal{S}_{\mathcal{G}}$ is also positive. Figure 7a shows that $\mathcal{S}_{D_{KL}}$ and $\mathcal{S}_{\mathcal{G}}$ are close to each other, which means $\mathcal{S}_{\mathcal{G}}$ is a reasonable estimator. Also, when $\mathcal{S}_{\mathcal{G}}$ is negative, it is close to zero. It indicates that the estimation $\mathcal{S}_{\mathcal{G}}$ is reliable when large. So, a possible extension is to use a threshold to filter cases where $\mathcal{S}_{\mathcal{G}}$ are close to zero. Such cases also have small $\mathcal{S}_{D_{KL}}$, so we can understand it in the way that the cases are filtered because they are essentially hard to predict.

**Intervention** We also like to discuss the relation between intervention and $\mathcal{S}_\mathcal{G}$. Figure 7b shows $\mathcal{S}_\mathcal{G}$ can be negative only when $H(A)$ is negative. This means the prediction may not be robust when the intervention reduces the amount of information in the cause variable.

**Edge example** Based on the previous analysis, we show an example that $\mathcal{S}_\mathcal{G}$ is negative. The intervention reduces the entropy of the cause variable. $\mathcal{S}_\mathcal{G} = -0.086$

$$P_1(A) = \begin{bmatrix} 0.4 \\ 0.6 \end{bmatrix} \qquad P_2(A) = \begin{bmatrix} 0.2 \\ 0.8 \end{bmatrix} \qquad P(B|A=0) = \begin{bmatrix} 0.3 \\ 0.7 \end{bmatrix} \qquad P(B|A=1) = \begin{bmatrix} 0.6 \\ 0.4 \end{bmatrix}$$

**Details of the experiment** For each experiment, we generate the distributions in the following way. We first generate the invariant conditional distribution $P(B|A)$. We then generate two marginal distributions, $P_1(A)$ for training and $P_2(A)$ for transfer distribution. The joint distributions $P_2(A, B)$ and $P_2(A, B)$ are computed from the conditional and the marginal distributions. Each distribution is generated by drawing a number uniformly at random between 0 and 1 for each possible value and normalizing them.

