# OpenReview forum: "Efficiently Disentangle Causal Representations"
_CPAL.cc/2024/Conference — CPAL 2024 (Proceedings Track) Oral_

### Official Review · Reviewer_oMj8 · 2023-10-05
**A novel approach that enhances an existing method, supported by both solid theoretical justification and empirical evaluation.**

**Rating:** 7
**Confidence:** 2

**Review:**

**Overview**

This paper proposes an efficient and theoretically motivated approach to simplify the technique of disentangling causal representations.

**Strength**
* The key idea of approximating the difference of conditional probabilities with models' generalization abilities is intuitively reasonable and further carefully approved with corner case well discussed.
* The empirical evaluation is solid as the experiments follow the most standard work, and the efficiency improvement of the proposed method is very pronounced.


**Question**
* Does the paper offer insights into the approximation error associated with using generalization abilities as a surrogate for the actual difference in conditional probabilities? Is there any provided intuition or established bounds?

---

### Official Review · Reviewer_sut8 · 2023-10-10
**Solid theoretical analysis**

**Rating:** 8
**Confidence:** 3

**Review:**

This paper studies efficient learning of disentangled causal representations by approximating conditional probability differences with generalization loss.

**Quality**
The paper is technically strong with solid theoretical analysis and extensive experiments.

**Clarity**
The writing is clear and easy to follow. The problem is well motivated, and the proposed approach is intuitively explained. Theoretical results clearly convey the mechanisms and efficiency benefits.

**Originality**
Approximating conditional divergence with generalization loss to identify causality is novel. This simplification enables direct leverage of standard ML workflows.

**Significance**
The significant efficiency improvements enable wider application of causal representation learning.

## Pros
1. Intuitive approximation using generalization loss avoids the adaptation process.
2. Theoretical analysis clearly explains mechanisms and advantages.
3. Empirical results strongly demonstrate efficiency benefits.
4. Enables straightforward integration with standard ML workflows.

## Cons
1. More intuition behind the generalization loss approximation could be useful.
2. Experiments on more complex real datasets could better showcase benefits.
3. More discussion on sensitivity and failure cases would be helpful.
4. Comparison with more baselines besides the one previous method is needed.
5. A broader impact discussion could be added.

---

### Official Review · Reviewer_fXyy · 2023-10-13

**Rating:** 5
**Confidence:** 2

**Review:**

**Summary:**.
The paper explores a new approach to concurrently disentangling causal variables and determining the causal relationship between them. It draws inspiration from the work of Bengio et al. (2020), but diverges by focusing on the generalization gap instead of employing a meta-objective and adaptation speed.

**Pros:**
•	The proposed method addresses both learning a disentangled representation of causal variables and discovering their causal direction.

**Cons:**
•	The evaluation is somewhat limited. Specifically, it remains uncertain how the method can be extended to more than two variables and larger benchmarking datasets. Additionally, it would be beneficial to see ablation studies on edge cases, particularly the impact of choosing conditional and marginal distributions on the condition of a small entropy gap in Proposition 2 (which lies at the core of the theoretical justification of the method)

**Detailed:**
The proposed method presents an interesting solution for acquiring both disentangled (causal) representations and the causal relationships between them. It builds upon the concept introduced by Bengio et al. (2020), but simplifies the computations to a comparison of the generalization gap. Consequently, it eliminates the necessity for defining connection-wise structural parameters and employing REINFORCE-based gradient estimators. However, I do have some concerns about this work.

Firstly, I find the evaluation (and hence the impact) of the paper limited. The method is designed solely for setups involving two observational variables, and it does not appear readily extensible to studies involving datasets with more variables. As a result, its usefulness in recovering causal directionality in large-scale practical applications is unclear. Furthermore, the only baseline considered is the work of Bengio et al. (2020). While this is a logical baseline given the similarities between both approaches, it's worth noting that the notion of meta-learning the structural parameters of a causal graph has also been explored in more expansive and efficient studies (e.g., [2] and [3]).

Secondly, if I comprehend correctly, according to Proposition 1 (and consequently 2), the method is capable of working only with interventions on the cause variable (i.e. the causal graph cannot change between the transfer and train distributions). Consequently, there are no assurances regarding its behavior when the transfer dataset is, in fact, an intervention on the effect, thereby breaking the dependence between observed variables. Simultaneously, according to Proposition 2, the difference in generalization gap is posited as a valid predictor if the delta entropy gap between variables B and A is "reasonably small." However, the authors only provide an intuition that such a statement should not be violated in real-world applications. The paper does not examine any real-world datasets or distributions (or their "approximations," as encountered, for instance, in BnLearn, since it is difficult to request a "real" dataset), nor does it conduct ablation studies on potential edge cases that could lead to the violation of the aforementioned argument. I believe the paper could benefit from a more comprehensive analysis from this perspective.

**Questions:** How is σ(γ) defined in this approach?

**References:**.
[1] Bengio, Yoshua, et al. "A meta-transfer objective for learning to disentangle causal mechanisms." arXiv preprint arXiv:1901.10912 (2019).
 [2] Ke, Nan Rosemary, et al. "Learning neural causal models from unknown interventions." arXiv preprint arXiv:1910.01075 (2019).
[3] Lippe, Phillip, Taco Cohen, and Efstratios Gavves. "Efficient neural causal discovery without acyclicity constraints." arXiv preprint arXiv:2107.10483 (2021).

---

### Meta-Review · Area_Chair_RnYz · 2023-11-13

**Recommendation:** Accept (Poster)
**Confidence:** 4

**Metareview:**

The reviewers are generally positive about the paper and find it technically strong, original, and well-written. Several concerns were raised, including the limitation to two observational variables and limited experimental evaluation. Overall, the paper has sufficient novel contributions and is worth acceptance.

---

### Decision · Program_Chairs · 2023-11-19

**Decision:**

Accept (Oral)

**Comment:**

This paper proposes a new approach to disentangling causal representations based on the difference of conditional probabilities in the original and new distributions, which is more sample-efficient and faster than prior art. The paper is worthy of acceptance.

The action PC chair for this paper is Yuejie Chi, who made the decision after carefully reading the paper as well as the comments by all reviewers and AC. The decision is agreed by all PC chairs.